# Diindolylmethane Derivatives: New Selective Blockers for T-Type Calcium Channels

**DOI:** 10.3390/membranes12080749

**Published:** 2022-07-30

**Authors:** Dan Wang, Pratik Neupane, Lotten Ragnarsson, Robert J. Capon, Richard J. Lewis

**Affiliations:** 1Division of Chemistry and Structural Biology, Institute for Molecular Bioscience, The University of Queensland, Brisbane, QLD 4072, Australia; danwang@ujs.edu.cn (D.W.); pratik.neupane.1984@gmail.com (P.N.); l.ragnarsson@imb.uq.edu.au (L.R.); r.capon@imb.uq.edu.au (R.J.C.); 2Department of Chinese Medicine and Pharmacy, School of Pharmacy, Jiangsu University, Zhenjiang 212013, China

**Keywords:** 3,3′-diindolylmethane, natural anticancer agent, T-type calcium channels, selective blockers

## Abstract

The natural product indole-3-carbinol (I3C) and its major digestive product 3,3′-diindolylmethane (DIM) have shown clinical promise in multiple forms of cancer including breast cancer. In this study, we explored the calcium channel activity of DIM, its synthetic derivative 3,3′-Diindolylmethanone (DIM-one) and related I3C and DIM-one analogs. For the first time, DIM, DIM-one and analog IX were identified as selective blockers for T-type Ca_V_3.3 (IC_50_s DIM 2.09 µM; DIM-one 9.07 µM) while compound IX inhibited both Ca_V_3.2 (6.68 µM) and Ca_V_3.3 (IC_50_ = 3.05 µM) using a FLIPR cell-based assay to measure inhibition of T-type calcium channel window current. Further characterization of DIM by electrophysiology revealed it inhibited inward Ca^2+^ current through Ca_V_3.1 (IC_50_ = 8.32 µM) and Ca_V_3.3 (IC_50_ = 9.63 µM), while IX partially blocked Ca_V_3.2 and Ca_V_3.3 inward Ca^2+^ current. In contrast, DIM-one preferentially blocked Ca_V_3.1 inward Ca^2+^ current (IC_50_ = 1.53 µM). The anti-proliferative activities of these compounds revealed that oxidation of the methylene group of DIM shifted the selectivity of DIMs from breast cancer cell line MCF-7 to colon cancer cell line HT-29.

## 1. Introduction

Indole-3-carbinol (I3C), a phytochemical found in Brassica vegetables, has long been recognized as a chemopreventive agent to intervene in the early precancerous changes in carcinogenesis [1,2]. As expected, 3,3′-diindolylmethane (DIM), the main natural occurring substance produced following digestion of indole-3-carbinol, has been extensively investigated as an anticancer agent in vivo and in vitro, and has undergone clinical trials for multiple forms of cancer, including prostate cancer (Phase II) (ClinicalTrials.gov Identifier: NCT00888654) and breast cancer [3,4]. Interestingly, DIM has synergistic anticancer effects with other clinically used drugs, including paclitaxel [5], tamoxifen [4,6,7], capsaicin [8,9], cisplatin [10] and genistein [11], which potentiate the effect of the traditional drugs and/or attenuate their side effects. Pharmacological studies have revealed that DIM and its analogs interact with distinct signaling pathways that have been targeted for anticancer therapies, including the epidermal growth factor receptor (EGFR) tyrosine kinase (inhibition) [12], Hippo (activation) [13], Akt and phosphatidylinositide 3-kinase (PI3K) (inhibition) [14], histone deacetylase (HDAC) (inhibition) [15], and signal transducer and activator of transcription 3 (STAT3) (inhibition) [10]. A recent study also identified DIM derivatives as potent agonists of the pathologic fibrotic pathway G protein-coupled receptor GPR84 [16].

T-type calcium channels (Ca_V_3.x) are recognized as a potential target for novel cancer therapies as they are aberrantly expressed in various cancer cells or tumors [17,18] and are implicated in cancer cell proliferation [19,20,21]. Our previous study identified two bisindole alkaloid analogs of marine fungal product pseudellone C as novel and selective T-type blockers [22], suggesting the indole moiety may contribute to Ca_V_3.x inhibition. In this work, through an efficient regio- and chemoselective Friedel–Crafts acylation of indole [23] combined with sulfonation [24], tosyl protection as well as benzyl protection, we achieved 3,3′-diindolylmethanone (DIM-one, II) along with nine analogs III–XI. To investigate their pharmacological potential, DIM, DIM-one and analogs III–XI, three I3C analogs included, were explored for activity on voltage-gated calcium channels (VGCC) using a FLIPR cell-based assay. For the first time, DIM, DIM-one, along with its analog 1 bis(1-benzyl-1H-indol-3-yl)methanone (IX) were revealed to be potent blockers of Ca_V_3.x window current, and were further characterized in whole-cell patch-clamp studies. In addition, DIM-one (II), III, IV and V showed anti-proliferative potential in colon cancer.

## 2. Materials and Methods

### 2.1. Chemical Materials

DIM was purchased from Sigma-Aldrich (St. Louis, MO, USA), DIM-one (II), and analogs III–XI were prepared synthetically. Synthetic procedures and NMR (DMSO-*d*_6_) data for each compound, see the Appendix A.

### 2.2. Cell Culture and Transient Expression

The human embryonic kidney 293 (HEK 293) cell lines (from Emmanuel Bourinet, Montpellier, France) stably expressing Ca_V_3.2 or Ca_V_3.3 were cultured under 5% carbon dioxide at 37 °C in Dulbecco’s Modified Eagle Medium (DMEM) Glutamax (Gibco, Life Technologies, Carlsbad, CA, USA) supplemented with 10% (*v/v*) fetal bovine serum (FBS), 100 U/mL penicillin, 100 μg/mL streptomycin (Gibco, Life Technologies, Carlsbad, CA, USA), and 750 μg/mL geneticin (G418) (Gibco, Life Technologies). The Chinese Hamster Ovary (CHO) cell lines (Emmanuel Bourinet, Montpellier, France) expressing Ca_V_3.1 were cultured under 5% carbon dioxide at 37 °C in Alpha Minimum Essential Media (MEM) Glutamax (Gibco, Life Technologies), supplemented with 10% (*v/v*) fetal bovine serum (FBS) and 300 μg/mL geneticin (G418) (Gibco, Life Technologies). The human neuroblastoma SH-SY5Y cells (Victor Diaz, Goettingen, Germany) were cultured under 5% carbon dioxide at 37 °C in RPMI 1640 antibiotic-free medium (Invitrogen, Carlsbad, CA, USA), supplemented with 15% FBS and 2 mM GlutaMAX™ (Invitrogen). D-PBS (Gibco, Life Technologies) was used to wash the cells, and 0.25% Trypsin-EDTA (Gibco, Life Technologies) was used to detach cells from the flask surface. They were split in a ratio of 1:5 (ideally 1000 cells/cm^2^) when they reached 70–80% confluence (every 2–3 days). Transiently transfected Ca_V_3.1 HEK293 T cells were used in FLIPR Assays. HEK 293 T cells were cultured under 5% carbon dioxide at 37 °C in DMEM supplemented with 10% (*v/v*) FBS. D-PBS was used to wash the cells, and 0.25% Trypsin-EDTA was used to detach cells from the flask surface. The cells were split and seeded at 6 million cells per T175 flask, to reach 70–80% confluence after 24 h. The next day, 12 μg DNA of human Ca_V_3.1 was incubated in 900 μL serum-free DMEM with 36 μL FuGENE HD transfection reagent (Promega Corporation, Madison, WI, USA) (1:3 DNA/Fugene ratio) for 20 min, and then the mixture was added into the cell flask slowly, drop by drop. After the transfection, the cells were cultured under 5% carbon dioxide at 37 °C for 24 h and then moved to a 28 °C incubator.

Human breast cancer cell line MCF-7 and T-47D, lung cancer cell line A549, and colon cancer cell line HT-29 were cultured under 5% carbon dioxide at 37 °C in DMEM Glutamax (Gibco, Life Technologies) supplemented with 10% (*v*/*v*) FBS. They were split in a ratio of 1:3–1:6 (ideally 2–4 × 10,000 cells/cm^2^) when they reached 70–80% confluence using 0.25% Trypsin-EDTA. For cell viability MTT assay, cells were seeded into 96-well clear bottom culture plates (Corning, Lowell, MA, USA) at a density of 5 × 10^3^ cells/well and settled at 37 °C for 24 h.

### 2.3. T-Type Calcium Channel Window Current FLIPR Assays

HEK 293 cells stably expressing Ca_V_3.2 or Ca_V_3.3 were seeded into 384-well black wall clear bottom plates (Corning) at a density of 30,000 cells per well. Transiently transfected Ca_V_3.1HEK293 T cells were seeded into 384-well black wall clear bottom plates at a density of 60,000 cells per well. Once the cells reached 90–95% confluence after 24 h, the media were removed from the wells and replaced with 20 μL of 10% calcium 4 dye (Molecular Devices, Sunnyvale, CA, USA) in HBSS-HEPES (containing 5 mM KCl, 10 mM HEPES, 140 mM NaCl, 10 mM glucose, and 0.5 mM CaCl_2_, pH 7.4) with 0.1% bovine serum albumin (BSA). The cells were incubated for 30 min at 37 °C in the presence of 5% carbon dioxide. Each well on the reagent plates for the first addition contained 15 μL different concentrations of compounds dissolved in HBSS-HEPES containing 0.1% BSA and <0.5% DMSO and was incubated for 20 min after loaded. Positive and negative controls contained 15 μL of HBSS-HEPES (0.1% BSA) alone. The plates were placed in the FLIPR^TETRA^ (Molecular Devices, Sunnyvale, CA, USA) programmed to measure maximum fluorescence intensity following a second addition of the agonist 5 mM CaCl_2_. The fluorescence readings were recorded and converted as described previously [22], and HBSS-HEPES (0.1% BSA) was used in the second addition as a negative control.

### 2.4. HVA Calcium Channel FLIPR Assays

SH-SY5Y cells were seeded into 384-well black wall clear bottom plates at a density of 15,000 cells per well, resulting in 90–95% confluence after 24 h. The media were then removed from the wells and replaced with 20 μL of 10% calcium 4 dye (Molecular Devices) in physiological salt solution (PSS) (containing 5.9 mM KCl, 1.4 mM MgCl_2_, 10 mM HEPES, 1.2 mM NaH_2_ PO_4_, 5 mM NaHCO_3_, 140 mM NaCl, 11.5 mM glucose, and 1.8 mM CaCl_2_, pH 7.4) with 0.1% BSA. As reported [25], for N-type calcium channel FLIPR assays the cells were pre-incubated with 10 µM nifedipine added in the dye to ensure full inhibition of L-type calcium responses. For L-type calcium channel FLIPR assays, the cells were pre-incubated with 1 µM CVID added in the dye to ensure full inhibition of N-type calcium responses. Positive control on the first reagent plate contained 15 μL of PSS (0.1% BSA), whereas PSS (0.1% BSA) containing 1 µM CVID and 10 µM nifedipine (final concentration) was used as a negative control. The fluorescence readings were recorded and converted as described previously [25], and agonist containing 90 mM KCl + 5 mM CaCl_2_ was used in the second addition.

### 2.5. Whole-Cell Patch-Clamp Electrophysiology

Whole-cell patch-clamp experiments were performed on an automated electrophysiology platform QPatch 16 X (Sophion Bioscience A/S, Ballerup, Denmark) in single-hole configuration using 16-channel planar patch chip QPlates (Sophion Bioscience A/S). The extracellular recording solution contained, in mM: TEACl 157, MgCl_2_ 0.5, CaCl_2_ 5, and HEPES 10; pH 7.4 adjusted with TEAOH; and osmolarity 320 mOsm. The intracellular pipette solution contained, in mM: CsF 140, EGTA 1, HEPES 10, and NaCl 10; pH 7.2 adjusted with CsOH; and osmolarity 325 mOsm. Compounds were diluted in extracellular recording solution with 0.1% BSA at the concentrations stated (DMSO ≤ 0.1%), and the effects of compounds were compared to the control (extracellular solution with 0.1% BSA) parameters within the same cell. Compounds’ incubation time varied from two (for the highest concentration) to five (for the lowest concentration) minutes by applying the voltage protocol 10–30 times at 10 s intervals to ensure steady-state inhibition was achieved. The effects of compounds were obtained using 200 ms voltage steps to peak potential from a holding potential of −90 mV. Current–voltage (*I*–*V*) relationships were obtained by holding the cells at a potential of −100 mV before applying 50 ms pulses to potentials from −75 to +50 mV every 5 s in 5 mV increments. Data were fitted with a single Boltzmann distribution: *I*/*I*_max_ = {1 + exp[*V* − *V*_5_/*k*}^−1^, where *V*_50_ is the half-availability voltage and *k* is the slope factor. Off-line data analysis was performed using QPatch Assay Software v5.6 (Sophion Bioscience A/S) and Excel 2013 (Microsoft Corporation, Redmond, WA, USA).

### 2.6. Cell Viability MTT Assay

Seeded cells were treated with various concentrations of compounds for desired time period (24, 48 and 72 h). After the treatments, the media were removed from each well, and replaced with 25 μL of serum-free media and 25 μL of MTT Reagent (cat. no., ab211091; Abcam, Cambridge, MA, USA). The plate was then incubated at 37 °C for 3 h, and 75 μL of MTT Solvent (cat. no., ab211091; Abcam) was added into each well after incubation. The absorbance was evaluated at 590 nm. Cell-wells treated with 0.1% DMSO were used as positive control and no cell-wells were used as background control.

### 2.7. Data Analysis

Data were plotted and analyzed using GraphPad Prism v7.0 (GraphPad Software Inc., San Diego, CA, USA). A four-parameter logistic Hill equation with variable Hill coefficients was fitted to the data for concentration-response curves. Data are means ± SEM of *n* independent experiments. Statistical analysis was performed with Two-way analysis of variance (ANOVA) with statistical significance at *p* < 0.05.

## 3. Results

### 3.1. Synthesis of 3,3′-Diindolylmethanone and Related Analogs

The synthesis of DIM-one (II) (60% yield) was achieved by Friedel–Crafts acylation of indole, using 1*H*-indole-3-carboxylic acid in dry dichloroethane (DCE) in the presence of ZrCl_4_, a method first reported by Guchhait et al. [23]. To better explore the structure–activity relationship (SAR) of the synthetic DIM analogs on VGCC, sulfonation [24,26] has been applied to achieve analogs III and V (revised structure of the rare sponge metabolite echinosulfone A [26]) (Figure 1), and tosyl and benzyl protection [26] have been applied to achieve analogs IX–XI (Figure 1), resulted in symmetric and asymmetric substitutions on the indole rings. Three I3C analogs VI–VIII were also made, and VIII, previously reported with weak pyruvate kinase inhibitory activities [27], was achieved by Friedel–Crafts acylation of indole and oxalyl chloride.

### 3.2. Evaluation of VGCC Activities of the Synthetic Compounds Using FLIPR Cell-Based Assays

DIM (I), DIM-one (II), and its analogs III–XI were evaluated for activity on VGCC using FLIPR cell-based assays. Their IC_50_ values as well as structures were summarized in Table 1. The mono-indole VII showed moderate inhibition against Ca_V_3.2 and Ca_V_3.3 current measured in T-type window current assays with similar IC_50_s of 17.07 ± 1.87 µM (*n* = 3) and 13.84 ± 2.02 µM (*n* = 3), respectively. Among the tested bis-indole compounds, VIII is highly cytotoxic, which caused abnormal current in cells with aberrant partterns, while XI is highly insoluble in buffer with 0.1% BSA. III showed moderate inhibition against Ca_V_3.3, while compounds IV, V, VI, and X showed poor inhibition of all the Ca^2+^ responses. In contrast, DIM, DIM-one and IX were identified to be selective Ca_V_3.x blockers with good potency.

DIM had the best potency and selectivity against Ca_V_3.3 window current, with an IC_50_ value of 2.09 ± 0.43 µM (*n* = 3), which was >27-fold better than its potency at high voltage-activated (HVA) Ca_V_s, and ~25-fold better potency at Ca_V_3.1 and Ca_V_3.2. Comparatively, DIM-one, which has an oxidized methylene group, showed a 4.3-fold reduced potency for Ca_V_3.3 window current with an IC_50_ value of 9.07 ± 0.69 µM (*n* = 3) and a ~1.4-fold reduced potency for Ca_V_3.2 window current compared to DIM, with an IC_50_ value of 73.85 ± 2.48 µM (*n* = 3).

DIM-one analog, IX potently blocked both Ca_V_3.2 and Ca_V_3.3 responses with IC_50_ values of 3.05 ± 0.51 µM (*n* = 3) and 6.68 ± 0.79 µM (*n* = 3), respectively, while it was >7-fold less active at high voltage-activated (HVA) Ca_V_s. In general, compounds with electron-donating substituents had better Ca_V_3.x activity, while electron-withdrawing substituents compromised activity. The fluorescent Ca_V_3.2 and Ca_V_3.3 Ca^2+^ responses before and after addition of DIM, DIM-one and IX, and their representative concentration-response curves, are presented in Figure 2 (Ca_V_3.2) and Figure 3 (Ca_V_3.3), respectively.

### 3.3. Electrophysiological Characterization of the Selective Ca_V_3.x Blockers in QPatch Assays

We also examined the effects of DIM, DIM-one and IX on the Ca_V_3.x by whole-cell patch-clamp using the automated electrophysiology platform QPatch 16 X (Figure 4, Figure 5 and Figure 6), and their IC_50_s were summarized in Table 2. DIM modestly inhibitedCa_V_3.2 and Ca_V_3.3 whole-cell current, with IC_50_ values of 21.09 ± 1.19 µM (*n* = 4) and 9.63 ± 0.97 µM (*n* = 5), respectively, and slightly better potency against Ca_V_3.1 whole-cell current, with an IC_50_ value of 8.32 ± 1.53 µM (*n* = 5). Compared to DIM, DIM-one had poor inhibition of Ca_V_3.2 and Ca_V_3.3 whole-cell current but potently inhibited Ca_V_3.1 whole-cell current with an IC_50_ value of 1.53 ± 1.06 µM (*n* = 3). Interestingly, low concentrations of DIM (49 nM–3.1 µM) weakly enhanced Ca_V_3.1 channel current before inhibiting current at higher concentrations, without associated changes in current characteristics. Biphasic effects found in drugs, e.g., the chemotherapy medication doxorubicin [28] and the psychoactive cannabinoids THC [29] and CBD [30] are crucial in determining dosage. Similarly, this biphasic effect of DIM in Ca_V_3.1 could potentially aid the DIM dosing in future clinical trials, although further functional studies are still needed. Current–voltage (*I–V*) relationships of Ca_V_3.1 in the presence of low (2.5 µM) and high (15 µM) concentrations of DIM revealed that block was not accompanied by shifts in the *I–V* relationship (Figure 7).

### 3.4. Effects of the Synthetic Compounds on Cancer Cell Viability in MTT Assay

DIM has been investigated extensively for anticancer activity in vitro and in vivo. However, the diindolemethanones, including DIM-one, have not been reported for anticancer activity previously. Based on our initial tests (data not shown), DIM and related analogs II–X were tested at 50 µM on the human breast cancer cell lines MCF-7 (Ca_V_3.2 enhanced) and T-47D (Ca_V_3.2 and Ca_V_3.3 enhanced), the lung cancer cell line A549 (Ca_V_3.1 and Ca_V_3.2 enhanced), and the colon cancer cell line HT-29 (No enhanced expression of Ca_V_3.x) (Ca_V_3.x status see the Human Protein Atlas: https://www.proteinatlas.org/, accessed on 16 July 2022). The anti-proliferative activities of the compounds are concluded in Figure 8. DIM had the strongest anti-proliferative activities on MCF-7 breast cancer cell line as previously found. In contrast, DIM-one (II), III, IV and V showed preferential anti-proliferative activity on the HT-29 colon cancer cell line, which does not have enhanced expression of Ca_V_3.x. These data suggest the anti-proliferative activity of these compounds is unrelated to Ca_V_3.x inhibition. Interestingly, compound IX had significant activity at Ca_V_3.2 and Ca_V_3.3 window currents, but did not affect cancer cell viability, whereas Compound III promoted A549 cell proliferation but reduced HT-29 cell viability. In future studies, the involvement of other cancer-related signaling pathways should be explored for these analogs.

Based on the promising data obtained from the one-concentration antiproliferative effects, we decided to examine the concentration response effects of DIM, DIM-one, IV and V. The concentration-dependent effects of the compounds on the four cancer cell lines after 72 h treatment are shown in Figure 9.

The MTT assay results showed that DIM had similar antiproliferative effects on breast cancer cell lines MCF-7 and T-27D and the colon cancer cell line HT-29, but was slightly less effective on the lung cancer cell line A549. Comparatively, all three methanone (DIM-one) compounds showed preferential antiproliferative effects on the non-Ca_V_3.x enhanced colon cancer cell line HT-29 compared with the other three Ca_V_3.x enhanced cancer cell lines. DIM-one and V showed ≥2 fold better antiproliferative effect on HT-29 cell viability than their effects on the breast cancer cell lines and lung cancer cell line A549, whereas IV was ~3 fold more effective on HT-29 cell viability compared with its effects on the other three cancer cell lines. Compound IV also showed >10 fold improved potency on HT-29 cell viability compared with the other three compounds. Further investigations revealed that instead of a continually increasing effectiveness with elongated treatment times, 48 h and 72 h treatment of IV showed similar antiproliferative effects on all four cell lines (see the Appendix A).

## 4. Discussion

In this study, we demonstrated that the natural product DIM, along with its synthetic derivative DIM-one and IX, are potent and selective Ca_V_3.x current blockers that showed preferential inhibition of Ca_V_3.2 (IX) and/or Ca_V_3.3 (DIM, DIM-one, and IX) window current measured in FLIPR assays, while DIM-one showed preferential inhibition of Ca_V_3.1 whole-cell current QPatch assays. Unlike QPatch assays, where cells were held in hyperpolarized potentials, the FLIPR window current assay applied to Ca_V_3.x was permissive of a window current resulting from incomplete inactivation [31], where most of the channels are in inactivated state. Inhibition of Ca_V_3.x window current has been proposed to be privileged target for the development of new analgesic, antiepileptic, and anticancer drugs [32,33,34]. However, our data indicated that the anti-proliferative activities of the investigated compounds against cancer cells are unrelated to their Ca_V_3.x inhibition.

As mentioned above, IX fully blocked Ca_V_3.2 and Ca_V_3.3 current with good potency in the FLIPR window current assay, but to our surprise, they showed only partial inhibition of both Ca_V_3.2 and Ca_V_3.3 whole-cell current in QPatch assays, where affinity is determined by interactions with the resting state of the channel. Both DIM and DIM-one showed good selectivity for Ca_V_3.3 window current, while in QPatch assays, DIM showed good potency for both Ca_V_3.1 and Ca_V_3.3 activated current and DIM-one showed preferential inhibition of Ca_V_3.1 activated current with an IC_50_ value of 1.53 ± 1.06 µM (*n* = 3). Interestingly, low concentrations of DIM (49 nM–3.125 µM) have been observed to show a promoting effect of Ca_V_3.1 channel opening, and started to show inhibitory effects with higher concentrations. However, neither low nor high concentrations of DIM showed voltage-dependent effect on *I–V* relationships of Ca_V_3.1. Given overexpression and gene knockdown of Ca_V_3.1 decrease MCF-7 cell proliferation [20,35], DIM may help clarify the role(s) of Ca_V_3.1 in MCF-7 cell proliferation.

T-type Ca_V_3.1 and Ca_V_3.2 have been reported to play critical roles in neurological disorders and diseases like absence epilepsy [36,37,38,39], inflammatory pain [40], etc. As a rising anticancer agent under several clinical trials, DIM has been reported to show synergistic anti-colorectal cancer effect with capsaicin [8,9], which is an analgesic agent for peripheral nerve pain [41]. Combined with this work, the Ca_V_3.x inhibitory activity of DIM may indicate its possible application in pain relief.

DIM has been explored for clinical application mainly in breast cancer therapy. In this work, cell viability MTT assay has also revealed that DIM has a preferable anti-proliferative activity on MCF-7 breast cancer cell line, while DIM-one along with its analogs III, IV, and V showed better anti-proliferative activity on HT-29 colon cancer cell line. Among them, IV and V also showed promising anti-proliferative activities on MCF-7, T-47D and A549 cell lines, which requires further studies to determine their targeting signaling pathways.

In summary, for the first time, the natural anticancer product DIM, along with its synthetic derivatives DIM-one and IX were characterized as promising and selective Ca_V_3.x blockers, which could possibly guide a broader application of DIMs in clinical use. In general, the oxidation of DIM methylene group compromised its Ca_V_3.x activities. However, these DIM derivatives, including DIM-one, III, IV and V, showed preferable anti-proliferative activities on non-Ca_V_3.x enhanced HT-29 cell line, highlighting their potential as early leads in the development of new colon cancer therapies.

## Data Availability

Data are contained within article.

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
