# Peer review of "Diindolylmethane Derivatives: New Selective Blockers for T-Type Calcium Channels"

_membranes, 2022, doi:10.3390/membranes12080749_

Round 1

Reviewer 1 Report

No further comments

Author Response

No further changes requested.

Reviewer 2 Report

The manuscript describes explores the calcium channel activity of DIM and its analogs. In general, the manuscript presents interesting data and is well-written.

I have the following suggestions and concerns about this manuscript.

1. Small language corrections are needed e.g.: add a comma in appropriate places, line 205: add “-“ in concentration response curve, line 189: identified instead indentifide – I recommend checking carefully manuscript

2. The compounds were well characterized. Nevertheless, the description in the experimental part requires some corrections. I recommend using one style of description of NMR in a table or in the text. In chemistry (Supplementary data) is no description of a style of NMR- I recommend adding information for HNMR (600 HZ) and CNMR (150 Hz).

3. In section 3.4 “Effects of synthetic …”  and in the Supplementary data are the same description and figures. I recommend removing text and figures from Supplementary data and leaving them only in the main text.

4. For MTT  assay authors studied activity only against cancer cell lines. I recommend adding results also for a normal cell line.

5. The majority of figures (Fig 2-7, Fig 9) in the main text (as well as in supplementary data) seem to be hardly legible and difficult to analyze. I recommend enlarging the figures or changing the resolution of the drawings.

Author Response

The manuscript describes explores the calcium channel activity of DIM and its analogs. In general, the manuscript presents interesting data and is well-written. I have the following suggestions and concerns about this manuscript.

  1. Small language corrections are needed e.g.: add a comma in appropriate places, line 205: add “-“ in concentration response curve, line 189: identified instead indentifide – I recommend checking carefully manuscript

Response: We thank the reviewer for identifying the errors which have been corrected as requested, and grammatical edits are made throughout the manuscript using “Track Changes”.

  1. The compounds were well characterized. Nevertheless, the description in the experimental part requires some corrections. I recommend using one style of description of NMR in a table or in the text. In chemistry (Supplementary data) is no description of a style of NMR- I recommend adding information for HNMR (600 HZ) and CNMR (150 Hz).

Response: We thank the reviewer for the comment. The tables were to compare similar structures using correlation of proton and carbon, and to confirm structures. Considering that it is not the focus of this work, we have now changed the Tables into text to make consistent descriptions, and we have added a sentence of “1H NMR was obtained in 600 Hz, and 13C NMR was obtained in 150 Hz.” in the first paragraph of Supplementary Materials.

  1. In section 3.4 “Effects of synthetic …”  and in the Supplementary data are the same description and figures. I recommend removing text and figures from Supplementary data and leaving them only in the main text.

Response: We appreciate the reviewer’s insight. We have now removed the repeated information from Supplementary Material.

  1. For MTT  assay authors studied activity only against cancer cell lines. I recommend adding results also for a normal cell line.

Response: Thank you for the comment. The purpose of the MTT assay was to have an initial idea of whether different compounds have different antiproliferative effects over different cancer cell lines. The suggestion of comparing the effect of the compounds on corresponding normal cell lines is beyond the scope of the current manuscript but should be considered in future toxicological investigations.

  1. The majority of figures (Fig 2-7, Fig 9) in the main text (as well as in supplementary data) seem to be hardly legible and difficult to analyze. I recommend enlarging the figures or changing the resolution of the drawings.

Response: We thank the reviewer for the comment. The figures are now enlarged within the layout. For resolution, we have used 1200 dpi, which we believe meet the journal requirements.

This manuscript is a resubmission of an earlier submission. The following is a list of the peer review reports and author responses from that submission.

Round 1

Reviewer 1 Report

The authors describe the preparation and analysis of 10 oxidized diindolylmethane and diindolylethane derivatives as blockers of T-type calcium channels and cytotoxic agents against four cancer cell lines. Several of the structures show modest inhibition of select T-type calcium channels (i.e., low mM IC50) and demonstrate cancer cell cytotoxicity at relatively high doses (50 mM) after three days. The authors do not establish whether the cytotoxicity is dose-dependent nor whether it is restricted to cancer cells. In fact, healthy cell lines were not evaluated for cytotoxicity suggesting the active compounds might be generally cytotoxic. In addition, evidently neither positive nor negative controls were used in any of the activity studies.

The structures for compounds VI and VII are incorrect in Scheme 1. The scheme for the preparation of VIII is also incorrect (i.e., VIII is not prepared from the indole-3-carboxylic acid, as shown). To top this off, the experimental procedure for the synthesis of VIII (1.7 in Supplementary) cannot work as described. One would have to start with indole for the procedure to have any chance for indicated product formation. Compound VIII was also previously reported as a species-selective pyruvate kinase inhibitor (BMCL 2014) and should be referenced as such. The acronyms ‘ETCOX’ and ‘COX’ in the Scheme 1 footer are rare (the former) or incorrect (the latter), and should be defined and corrected, respectively.

The authors repeatedly refer to compound XI, but there is no compound XI (only 10 compounds were studied). How can compound XI be highly insoluble in buffer [line 181] when there is no compound XI? The authors fail to describe the preparation of VI and VII (which are wrong in Scheme 1 to make matters even more confusing). Line 272 is confusing as written and line 305 should explicitly indicate and reference the other examples. The concluding statement, “which possibly make them good drug candidates for new colon cancer therapies [line 322-323]” is misleading. The low potencies, lack of toxicity data, and lack of target selectivity data would at best qualify the most active structures as hits or early leads. Claiming these are potential drug candidate compounds for colon cancer is not appropriate.

In summary, the manuscript is poorly edited and peppered with confusing errors. In addition, the bioactivity studies lacked control experiments, and no healthy cell toxicity data was reported, calling to question the significance of all the cell viability studies where activity was observed. Some experimental procedures are irreproducible as reported, and no spectra (e.g., NMR or HPLC) are offered to support compound purity. The title should also state ‘Diindolylethane’, since that framework comprises three of the studied derivatives. Given all these issues, several related to experimental viability and reproducibility, I do not believe this manuscript warrants publication.

Reviewer 2 Report

The presented analyzes, although interesting, are only preliminary and review studies. The authors did not check the cytotoxicity (MTT assay is not a good test to determine cytotoxicity), and no studies on the apoptotic or necrotic potential of the test compound have been included. There are no animal studies that could show in vivo activity as well as biodistribution of the analyzed substance. Unfortunately, the presented experiments are too sparse to consider publishing these results in a highly influenced journal.

Reviewer 3 Report

In their manuscript entitled “Diindolylmethane Derivatives: New Blockers for T-type Calcium Channels with Anticancer Activities”, Wang et al present interesting data suggesting that DIM and its derivatives could represent a new family of Cav3 inhibitors on top of exhibiting potential anticancer activities. Overall, this manuscript is clearly presented, and should prove of interest to a broad readership working on DIM.

Major comments:

Could the authors please clarify the relationship between Figures 1 and 2, and the Table 1? Indeed, values in the figures differ from the ones in the table, and are indicated without SEM despite the mention of 4 experimental repeats for each value.

Fig 6: the weak current enhancement mentioned in the text (lines 225-229) and in the discussion section by the authors is not visible on the graph. Please amend the text, or show a figure better illustrating this point.

Figure 7: have the authors checked the effect of their compounds in MCF7, T47-D, and A549 cells that do not overexpress Cav channels? Direct comparison between a same cell line overexpressing or not Cav channels would be more relevant here, and would greatly help assessing the potential clinical interest of these compounds. Moreover, the use of a MTT assay is problematic here as values could be greatly disturbed by the impact of the compounds on mitochondrial activity. Indeed, variations could originate from the inhibition of intracellular calcium entry, resulting in the down-regulation of mitochondrial metabolism without touching cell viability. Other methods (cell counting, flow cytometry, TUNEL…) are required before any conclusion can be drawn regarding the potential impact of the compounds on cell viability. At the very least, the text should be amended to reflect these potential issues.

Minor comment:

Lines 178-179: “µM” is missing from both numerical values.